# Continuous Assessment of the Environmental Impact and Economic Viability of Decarbonization Improvements in Cement Production

Olurotimi Oguntola and Steven Simske *

Systems Engineering Department, Colorado State University, Fort Collins, CO 80523, USA;
timi.oguntola@colostate.edu
* Correspondence: steve.simske@colostate.edu; Tel.: +1-970-241-5692

**Abstract:** Growing awareness of the importance of mitigating climate change is driving research efforts toward developing economically viable technologies for reducing greenhouse gas emissions. The high energy consumption and carbon-intensive nature of cement manufacturing make it worthwhile to examine the environmental and economic characteristics of process improvements in cement production. This study examines the environmental impact of cement production and its economic considerations and demonstrates an IoT-inspired deployment framework for continuously assessing these. It contributes a practical approach to integrating sustainability into cement manufacturing and analyzes four different scenarios from a combination of two cement types (ordinary Portland cement, Portland-limestone cement) and two energy sources for thermal heating (coal, dried biosolids). It indicates that increased production and adoption of blended cement that has up to 15% limestone as an alternative to ordinary Portland cement can significantly reduce climate change effects from cement production (6.4% lower carbon footprint). In addition, significant emission reduction is possible with the use of waste from sewage sludge as a combustion fuel for heating in the cement production process (7.9% reduction compared with baseline). The information on environmental and financial trade-offs helps informed decisions on cement production improvements and can potentially contribute to greenhouse gas reduction targets.

**Keywords:** cement manufacturing; life cycle assessment; techno-economic assessment; alternative inputs; continuous assessment deployment framework

## 1. Introduction

Climate change and further warming due to greenhouse gases in the atmosphere can significantly impact life. Its effects include frequent and severe storms, wildfires, floods, heavy precipitation in some regions and severe drought in others, a rise in sea level, and impacts on health, food, and water supply. In addition, concrete is a significant manufactured material whose main component is cement; therefore, monitoring the environmental implications of cement manufacturing is vital. The production of cement is estimated to be responsible for approximately 8% of the global carbon dioxide ($CO_2$) emissions caused by humans [1]; thus, emission reductions in cement manufacturing may lead to a decrease in greenhouse gas (GHG) emissions. Reports from major Western cement manufacturers such as LafargeHolcim and Heidelberg Cement indicate that more than 550 kg of $CO_2$ is emitted per ton of cement produced. However, due to the high temperatures required for its production and the direct emissions from the calcination of limestone, decarbonizing cement production is challenging and has been the subject of many research efforts [2–4].

Cement production has three stages: raw material extraction and preparation, clinker production, and cement grinding. First, limestone ($CaCO_3$) is ground with other minor constituents and heated at 900 °C via cyclones. The mixture is passed through the rotary kiln to produce a mixture of calcium silicates (cement clinker) from reactions at 1450–1500 °C [4].

The clinker is cooled, ground to a fine powder, and mixed with gypsum to produce cement. There are opportunities to mitigate $CO_2$ emissions from both process modification and energy efficiency because both process- and fuel-related emissions account for approximately 40% of total direct emissions. Some of the $CO_2$ mitigation methods suggested in the literature include carbon sequestration or carbon capture, utilization and storage (CCUS), use of alternative fuels in the kiln, energy recovery, waste heat recovery, and increasing the proportion of semi-dry and dry processes [5].

Cement companies can leverage carbon capture and storage technologies to capture $CO_2$ from significant point sources in their manufacturing process or the atmosphere, transport it, and permanently store it underground. The paper in [6] examined the limitations of deploying CCS technologies despite their availability and maturity. The authors advocate for expanding government policies to incentivize adopting CCS and mandate its deployment. Using alternative fuels properly can reduce the environmental impacts of the cement industry. Advancement in related research indicates that introducing solid waste materials as alternative fuels in cement manufacturing will lower energy consumption and reduce greenhouse gas emissions. By coupling the cement and waste management industries, solid waste materials such as municipal solid waste, sewage sludge, biomass, end-of-life tires, and meat and bone animal meal can be considered alternative fuels to replace or reduce the consumption of non-renewable fossil fuels in cement manufacturing [7].

More cement factories are now leveraging Waste Heat Recovery (WHR) systems in their bid to achieve energy performance as required by standards and legislation. The waste heat from the kiln is used as a power generation source, thus reducing thermal energy losses and improving the energy efficiency of the cement manufacturing process. In [8], the authors evaluated the performance of a WHR system, comparing it to the estimated performance from feasibility studies and proved positive financial indicators by comparing actual to updated capital expenditures. Energy efficiency can also be improved in the cement production process by leveraging process control and management systems, high-efficiency motors and drives, and efficient grinding technologies.

The process routes for the manufacture of cement are dry, semi-dry, semi-wet, and wet processes. The wet processes consume more energy than the dry processes. While the choice of process is primarily determined by the availability of raw materials, with expansion and significant improvements in the cement plant, semi-dry processes can be changed to dry processes to reduce GHG emissions [5].

Life cycle assessment (LCA) and techno-economic assessment (TEA) often are part of the validation for proposals of technologies aimed at reducing $CO_2$ emissions of industries, and carbon capture and utilization technologies are part of decarbonization options considered. When introducing new technology to reduce $CO_2$ emission, lower energy consumption, or capture and utilize carbon, it is imperative to perform LCA and TEA to ascertain the value added from the new technology or process change introduced. LCA aims to track the global environmental impacts of the production, use, and disposal of the product or service. TEA assesses the economic viability of the technology and is a tool for making decisions on research, development, investments, and policy.

Challenges with assessing technologies in their early stages result in an escalation of assessment efforts and potential mismatches of research results with the needs of stakeholders. To address this, [9] presented best practices for adapting assessment methodologies to the technology readiness level (TRL) of the technology when assessing early-stage climate change mitigation CCU technologies. The authors advocate for meeting stakeholders' needs by aligning TEA/LCA goals and scope with TRL rather than commercial interests, estimating missing data using standard estimation tools, and evaluating and communicating uncertainties in the assessment. They also recommend collaboration across technology developers and TEA/LCA practitioners as a workaround for coping with limited resources. With consensus on technological measures for decarbonization, industry watchers advocate for coupling effective policy with a body of research on technical solutions to cement and concrete decarbonization [10]. More decisive policy actions will help promote the adop-

tion of technological measures to decarbonize the cement industry. In addition, cement producers would benefit from a systematic way to continuously monitor and report the environmental impact of production processes to validate compliance with decarbonization policies as they adopt these technological measures.

This study examines the environmental impact of cement production and its economic considerations and demonstrates an IoT-inspired deployment framework for continuously assessing these. The LCA and TEA in this study analyze emission and cost reduction opportunities from alternative manufacturing inputs in four different scenarios at a United States cement plant. In addition, the study demonstrates a practical approach to integrating sustainability into cement manufacturing with a deployment framework for the continuous assessment of cement production's economic and environmental performance.

## 2. Materials and Methods

Digitalization and data-driven process optimization of cement manufacturing will help the industry better manage energy consumption and reduce emissions and raw material inefficiencies [11,12]. There has been improvement in sensors, data management, related Internet of Things (IoT) toolkits, and increased maturity levels of artificial intelligence. These, and the templatization of life cycle and techno-economic assessments, all contribute to a deployment framework for continuously assessing the economic and environmental impact of process improvements in cement manufacturing.

Advancing innovative near-zero emission production routes, and promoting material efficiency, are two of the key carbon-cutting strategies that would contribute the most to direct emission reductions in the Net Zero Scenario [13]. Near zero or net zero emission production of cement would require incorporating carbon capture in the production process, thereby increasing the cost of production. However, significant carbon emission reduction is possible by promoting material efficiency. This study explores material efficiency options in demonstrating the deployment framework using the Union Bridge, Maryland plant of Lehigh Cement (LC) as a case study. Heidelberg Cement of Germany wholly owns LC and has affiliations with technically advanced cement operations and construction-related materials activities. LC's original plant was built in 1910 and has since undergone several modernizations, including replacing four long-dry kilns with one preheater/pre-calciner kiln system. At the time of this study, the LC plant in Union Bridge, Maryland, is transitioning from producing ordinary Portland cement to Portland-limestone cement, which uses innovative technology to increase limestone content and reduce clinker used. According to the manufacturer, the product called EcoCem®PLC (Lehigh Cement, Union Bridge, MD, USA) contains as much as 10% more limestone but performs equivalent to ordinary Portland cement in terms of concrete compressive, flexural strength, and durability [14].

To demonstrate how the deployment framework can be leveraged for the continuous assessment and improvement of cement production's economic and environmental impact, we review its production at the LC plant in Union Bridge under four scenarios, as listed in Table 1 below.

**Table 1.** Cement production scenarios modeled.

|  | Scenario | Product | Thermal Energy |
|---|---|---|---|
| 1. | OPC + Coal | Ordinary Portland Cement | Coal |
| 2. | PLC + Coal | Portland-Limestone Cement | Coal |
| 3. | OPC + DBS | Ordinary Portland Cement | Dried Biosolids |
| 4. | PLC + DBS | Portland-Limestone Cement | Dried Biosolids |

In the first two scenarios, thermal energy for producing ordinary Portland cement and Portland-limestone cement is provided by coal combustion. In comparison, in the other two scenarios, thermal energy is provided by the combustion of dried biosolids from

processed sewage sludge. These scenarios were examined using the templates developed by the University of Michigan Global $CO_2$ Initiative [15].

*2.1. Life Cycle Assessment (LCA)*

The international standard ISO 14040 defined LCA as a study of environmental and other potential impacts throughout a product's life. The product's life, often called 'cradle-to-grave', includes raw material acquisition, production, use, and disposal. Environmental impacts include resource use, human health, and ecological consequences [16]. A detailed assessment of the whole life of a product that serves as an input in another product can be complicated. As a result, many researchers in practice limit the LCA to the use phase, often called 'cradle-to-gate'. The concept of LCA is based on a simplified system analysis. Therefore, meaningfully selecting and defining system boundaries are important albeit labor-intensive tasks within the LCA process. LCA can be applied to product development and improvement, public policy making, strategic planning, and marketing, amongst other direct applications. The main parts of the LCA are:

- Goal and scope definition (including functional unit and system boundaries);
- Life Cycle Inventory (LCI);
- Life Cycle Impact Assessment (LCIA);
- Interpretation.

As defined by the ISO standard 14040, the scope and goal of the LCA have to be clearly defined and consistent with the intended application. The inventory analysis involves compiling and quantifying inputs and outputs required throughout the product life cycle. The impact assessment component of the LCA aims to understand and evaluate potential environmental impacts throughout the product life cycle. In the interpretation phase, conclusions are drawn from the inventory analysis and impact assessment, and recommendations are made to satisfy the study's objective.

LCA of cement manufacturing has been the subject of many research efforts [17–19]. In addition, different localized research efforts focus on the environmental impact of cement manufacturing in different parts of the world, including India, Brazil, Europe, and China [20,21]. Leading international standards on LCA mainly focus on the process of performing LCA. Its principles and framework are described by ISO 14040, and ISO 14044 specifies requirements and provides guidelines [22].

Countries across the globe have also formulated a variety of standards and guidelines, such as the UK's PAS 2050 [23], France's BP X30-323, and Japan's EcoLeaf Environmental Labeling Program. However, the nature of a comprehensive life cycle analysis requires consideration of the inputs into the manufacturing process, which often differ from one location to another. It also requires consideration of the context of each manufacturing plant's infrastructure, processes, policies, and quality control requirements. As a result, the literature reviewed for LCA on cement manufacturing has varied in delivery, with each author articulating the environmental impact of cement production through differing lenses based on their goal of doing the analysis. As indicated in Table 2, LCA is used to study the environmental impact of different life cycle stages of cement production and usage, such as clinker, cement, mortar, and concrete.

In addition, the literature review identified other materials that can be used as additives in cement manufacturing or in the mix of mortar and concrete to reduce the carbon footprints of the products. These materials and the resulting estimated reduction in GHG emissions from their use include the following: marble waste sludges in cement −34% [24], ornamental stone waste in cement −9% [25], blast furnace fly ash and slag in concrete −32% [26], ash from wastewater treatment plant sludges in concrete −9% [27], plastic waste and carbon fibers in cement mortars −13.69% [28], and glass powder in cement mortar −20% [29].

**Table 2.** Parts of LCA and examples compiled from the literature.

| | Life cycle stage | Cement | Cement | Clinker | Clinker | Concrete | Concrete |
|---|---|---|---|---|---|---|---|
| Goal and scope definition | Functional unit | 1 ton Portland cement | 1 ton of ordinary Portland cement and 1 ton of clinker | 1 ton of clinker | 1 kg of clinker | Varied specific measures of concrete | 1 m$^3$ of concrete |
| | System boundaries | Raw materials and fuels extraction, transportation, electricity usage, and emissions | Life cycle inventory analysis | Cradle-to-gate LCA model. Clinker production in cement kiln, excluding blending and grinding | Cradle-to-gate LCA for old and new cement production lines. Clinker production, excluding blending and grinding | Modified cradle-to-gate. Comparison of traditional and 'green' concrete | Cradle-to-gate LCA of graphene production and use in concrete |
| | Country | Brazil | China | Switzerland | Spain | | UK |
| Life cycle Inventory (LCI) | Inputs and outputs | In—sand, limestone, clinker, chemical additives, and transportation Out—$NO_x$, $CO_2$, HCl, HF, Hg, Pb, Cd, Ta, and Dioxins | In—limestone, sandstone, ferrous tailings and gypsums, energy from coal and electricity, admixtures (fly ash and furnace slag, freshwater) Out—GHG, primary pollution, hazardous air pollutants, noise, heavy metal emissions | In—alternative fuel and raw materials (tires, prepared industrial waste, dried sewage sludge, blast furnace slag) Out—carbon, nitrogen, chloride, fluoride compounds, clinker, raw meal, cement, and kiln dust | In—limestone, sand, iron ore, clay, electricity generation, and heat Out—$CO_2$, $NO_x$, $SO_2$ particulates | In—minerals and fossil fuels, land use Out—$NO_x$, $SO_x$, $NH_3$, pesticides, heavy metals, $CO_2$, hydrochlorofluoro-carbons (HCFC), nuclides, polycyclic aromatic hydrocarbons (PAHs), volatile organic compounds, and suspended particulate matter (SPM) | In—Portland cement, ground granulated blast-furnace slag, limestone, sand, water, superplasticizer, graphene nanoplatelets paste, and input energy |
| | Data source | Plant, national statistics, and Ecoinvent database | On-site, 18 cement plants with 30 production lines from 2004 to 2007 | On-site, Ecoinvent database | On-site plant data. SimaPro 7.2 software. Ecoinvent 3.0 | LCA-related journals | Commercial companies and the scientific literature. SimaPro software |
| Life cycle Impact Assessment | LCIA method | | ISO Environmental Management—Life Cycle Assessment | Cumulative exergy demand (CExD) [30], eco-indicator | Cumulative exergy demand (CExD) [30] | IPCC 2007 Global Warming Potential (GWP) impact method | Impact 2002 + methodology [31] |
| | Impact analyzed | Ozone depletion, photochemical oxidant formation, terrestrial acidification, freshwater and marine eutrophication, and metal and fossil depletion | Freshwater consumption, noise emissions, heavy metal and hazardous pollution emissions, and indirect consumption of oil and coal | Gas emissions | Global warming, acidification, eutrophication, abiotic depletion, ozone layer depletion, freshwater aquatic ecotoxicity, and photochemical oxidation | Acidification, eutrophication, ecotoxicity, climate change, ozone layer depletion, ionizing radiation, respiratory effects, and carcinogenic | Carcinogens and non-carcinogens, respiratory inorganics, aquatic and terrestrial ecotoxicity, global warming, non-renewable energy, and mineral extraction |
| | Literature reference | [20] | [21] | [30] | [19] | [32] | [33] |

Practitioners agree that integrating LCA into day-to-day management routines will be beneficial; however, the execution of LCA is challenging. The LCA process can be kept simple without compromising comprehensiveness and reliability by using standard procedures and assumptions, adopting techniques that allow comparisons between different impact categories, access to high-quality data, and using adequate software [34]. Integrating LCA into management routines of cement plants has become feasible with international standards for LCA through the ISO process, increasing the number of cement plants getting more digitized and adopting IoT sensors for gathering data, and improved interconnectivity and access to databases through application programming interfaces (API). Section 2.3 demonstrates a theoretical framework for the continuous assessment of the environmental impact of cement production.

### 2.1.1. Scope of the Model: Functional Unit and System Boundaries

The functional unit adopted on a mass basis is the production of 1 metric ton of cement. The cradle-to-gate system boundary is used for this study. Only activities that occur before arriving at and within the cement plant are considered because performance and impacts after the cement plant are identical across product systems and irrelevant for impact comparison purposes. Figure 1 shows a representation of the system boundaries, system elements, and unit processes depicting the exchange of energy (E), particulate emissions (PE), gaseous emissions (GE), and heat (H) in the quarrying, crushing, grinding, dry mixing and blending, preheater, rotary kiln, clinker cooling, additives, and final grinding processes in cement production.

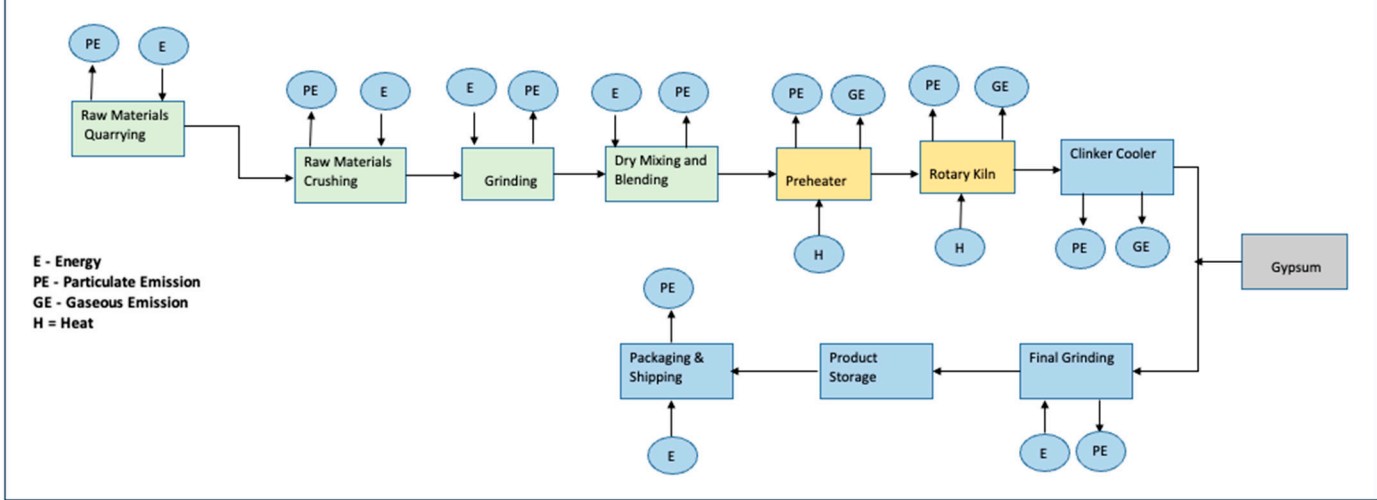

**Figure 1.** Graphical representation of the system boundaries, system elements, and unit processes.

### 2.1.2. Life Cycle Inventory

This is the compilation of information on the inputs, outputs, waste generated, electricity, and thermal energy required for the production of a functional unit of the product within the defined system boundary. The system boundary covers the following emission units listed by the Lehigh Cement Company in its operating permit as subject to Title V requirements and having applicable requirements [35]:

- Union Bridge quarry operations;
- New Windsor quarry operations;
- Raw material transport and storage;
- Raw grinding;
- Raw meal—kiln feed;
- Kiln and clinker cooler;
- Coal grinding mill for kiln;
- Clinker transport and storage;

- Clinker finish mills;
- Cement storage and shipping with bag packing;
- Dried-biosolids-related processes;
- Emergency generator.

The data switching for the scenarios for ordinary Portland cement, Portland-limestone cement, and the coal and alternative fuels are implemented using Excel's IF function on the cells of the inventory sheet highlighted in Figure 2 (also see the 'INVENTORY' sheet of the quantitative data spreadsheet). Relevant inventory data and impact assessment factors are sourced from the following—data on power consumption and expert opinion at the Union Bridge plant of Lehigh Cement; documentation from the parent company Heidelberg Materials; and calculation by coefficients and derivations from secondary data sources, including libraries and databases from the literature listed in notes amongst the sources listed in the 'SOURCE GUIDE' sheet of the quantitative data-sheet in the Supplementary Materials. Electricity from the national grid powers the crushing, grinding, conveying, and machine operation. Coal is used to generate the thermal energy required for the calcination process. This study also models production using thermal energy from the combustion of biosolids from treated sewage sludge from the Hampstead Waste Water Treatment Center in Carroll County, where the Lehigh Cement plant is situated. Dried biosolids are typically made from sewage sludge by maceration, pressurization, heating, decarboxylation reaction, and drying.

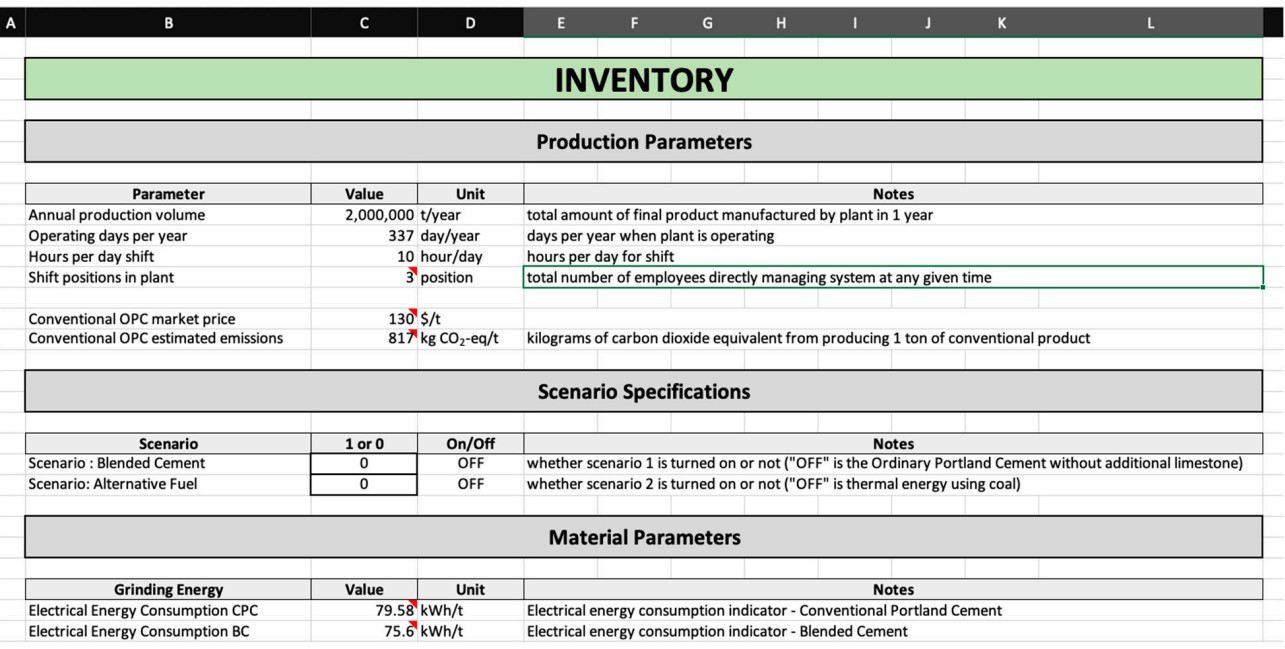

**Figure 2.** Subsection of the inventory sheet in the LCA.

2.1.3. Life Cycle Impact Modeling

The attributional LCA approach described in [36] was adopted for this demonstration. This approach does not analyze the indirect consequences of the product's manufacture. It is generally restricted to using average impact data to allocate the environmental impacts of factors in the product's life cycle stages. The impact assessment factors are limited to the climate change impact category considered for this study (see the quantitative data spreadsheet's 'Impact Assessment' sheet). Carbon dioxide ($CO_2$), methane ($CH_4$), and nitrous oxide ($N_2O$) are the relevant gases related to the greenhouse effect. However, the relative contribution of $CO_2$ predominates, being between 98.8% and 100%, because it is emitted in much higher quantities than the other gases [37]. Therefore, emissions from methane and nitrous oxide were excluded from this LCI.

*2.2. Techno-Economic Assessment*

The techno-economic assessment of the introduction of new technology or process is an important step when aiming to set a large-scale process, especially at the industrial level. To assess the economic viability of the change, TEA combines process modeling and engineering design with economic evaluation. TEA is apt for assessing decarbonization efforts from emission reduction methods in cement production.

System dynamics are considered suitable for handling the complexities of understanding the economic behavior of carbon capture and utilization (CCU) technologies. Apart from investments and operational costs, other factors such as government policies, market conditions, material and information delays, and the feedback process in the supply chain impact the economic behavior of CCU technologies [38]. In the referenced article, the authors simulated indirect carbonation using different hydroxides as absorbent precursors to reduce $CO_2$ emissions in clinker production. They performed an analysis of the $CO_2$ captured using a system dynamics model. They determined that $CO_2$ capture costs 65 to 140 USD/t$CO_2$ in the carbonation process and that a tax policy of 80 USD/t$CO_2$ or more will encourage the implementation of $CO_2$ capture.

In another assessment, the authors in [39] evaluated two Calcium Looping (CaL) processes for capturing $CO_2$ in cement plants. The first integrates the CaL in the cement kiln at the tail-end such that the placement of the $CO_2$ capture process is downstream in the clinker burning line and with fluidized bed reactors (CaO-rich sorbent). The other process integrates the CaL system with entrained flow reactors in which the carbonator is integrated with the preheater of the clinker burning line, treating only the flue gas from the rotary kiln. In their analysis, the authors determined that the tail-end and integrated $CO_2$ capture processes increased the cost of cement by 67% and 74%, respectively, while the cost of $CO_2$ avoided was 52 EUR/t$CO_2$ and 58.6 EUR t$CO_2$, respectively. Example metrics of emissions captured and the related cost of leveraging oxy-combustion with calcium looping vary from 94% of emissions captured at 17 USD/t$CO_2$ [40] to 60% of emissions captured at 40.6 USD/t$CO_2$ [41]. Table 3 summarizes decarbonization methods in the literature, the materials and equipment leveraged, and cost examples where available.

The following are instrumental to reducing cement prices and $CO_2$ emissions: carbon tax, $CO_2$ capture efficiency, cost-effective and energy-efficient amine blend, energy penalty, and $CO_2$ sales price [42]. Even with the opportunities for $CO_2$ capture from the process emissions from calcination, which by concession from the literature accounts for about 60% of cement production emissions, the cement industry is cautious about incorporating new technology that might affect clinker composition [43]. Therefore, alternative fuels with lower carbon footprints and technologies leading to lower energy requirements for heat generation for the kiln are also assessed for their economic viability at an industrial scale. The thermal energy (about 3.2–6.3 GJ per ton of clinker) required for cement production is provided by fossil fuels such as petroleum coke, natural gas, and coal. Alternative fuels considered in the literature for cement production include tire-derived fuels, commercial and industrial wastes, sewage sludge, meat and bone meal produced from slaughterhouse residue, agricultural biomass, and spent pot linings [44].

In a study exploring the economic feasibility of a waste heat recovery system that captures radiation emitted from the surface of a rotary kiln [45], the authors determined that for markets with electricity costing as much as 0.1 USD/kWh, the method could yield as much as 5% return on investment (ROI) or net present value of USD 0.06 million. The authors evaluated the system by combining computational fluid dynamics simulations with process modeling, including mass, energy, and exergy balances.

**Table 3.** Decarbonization methods.

| Method | Decarbonization Lever | Materials/ Equipment | Process Summary | Cost Example | Reference |
|---|---|---|---|---|---|
| **Methods Increasing Process Energy Efficiency—Process Decarbonization** | | | | | |
| Introduction of energy-efficient clinker technology with low cooling air requirement | Process decarbonization | Modern grate clinker coolers | Optimization of clinker coolers | Varies due to site specifics | [13] |
| Waste heat recovery | | Boiler/turbine system | Waste heat is used for drying, steam production, or feeding the local heat network. Decrease of 4–15 kg $CO_2$/t clinker | Depends on local power prices | |
| Replacing long wet/ semi-dry kilns with energy-efficient preheater/pre-calciner kilns | | Construction may be required | Raw material has lower moisture content. Additional cyclone stage. Thermal energy decrease of 900–2800 MJ/t clinker. Electrical energy decrease of 0–5 kWh/t clinker | A 35–50 M EUR investment and 2.85–9.2 EUR/t clinker decrease in operating cost | |
| **Methods utilizing alternative fuels—Circular Economy**, e.g., solid wastes, different biomass sorts, and fuels with lower heating values | Circular economy | Sewage sludge, wood waste, grain rejects, animal meal, mixed industrial waste, waste oil, tires, and plastics | Use for combustion in a pre-calciner vessel. Integrate waste management. Processing compliant with international environmental agreements and local policies | Investment costs for storage, handling, and pretreatment, lower operational costs, and 15–30% of coal price in Europe | [13] |
| **Methods utilizing different raw materials to reduce emissions from limestone decomposition—Circular Economy** | Circular Economy | Already decarbonated materials, e.g., metallurgical slags, coal ashes, and concrete crusher residues | Limits process-related and fuel-related $CO_2$ emissions | Limited availability of materials | [13] |
| **Decarbonization strategies—process decarbonization** | | | | | |
| Post-combustion capture. Decarbonizes flue gases generated from the total oxidation process | Process decarbonization | Solvents that react with $CO_2$, e.g., MDEA, MEA, DEA, AMP, and PZ * | $CO_2$ absorbing reaction, heat to reverse absorption, moisture removal, compression, transportation, and storage/utilization | 50.6 USD/ton [46] | [47] |
| | | Natural and synthetic calcium-based sorbents | | | |
| | | Polymeric membranes | Compress flue gas, pass through stages of membranes and compression to capture $CO_2$ | | [48] |
| Pre-combustion capture. Decarbonizes syngas resulting from fuel partial oxidation process before combustion | | Synthetic gas from feedstock (e.g., coal), steam, air, and heat | Water–gas shift reaction, $CO_2$ capture, separation, transportation, and sequestering | 60 USD/ton capture cost | [49] |
| Oxy-combustion uses oxygen rather than air for fuel total oxidation | | Oxygen-rich medium | Fuel combustion in a pure or enriched oxygen stream | 60–70 EUR/ton $CO_2$ avoided cost | [50] |
| **Other methods** | | | | | |
| Electrification and renewable procurement—clean energy | Clean energy | Synchronous power such as hydropower and biomass. Variable generation, such as wind and solar | Reusability, recyclability, and product longevity | Varies by site and is influenced by the price and availability of zero-carbon electricity | [51] |
| Eco planet and efficiency gains in construction—carbon-efficient Construction | Carbon-efficient Construction | Design and engineering techniques to reduce the amount of concrete required | Examples: curved fabric molds, pre-stressed concrete using tensioned steel cables. Concrete mixture optimization. | Varies | |

* MDEA—Methyl-Di-Ethanol-Amine; DEA—Di-Ethanol-Amine; AMP—2-Amino-2-Methyl-1-Propanol; PZ—Piperazine.

Techno-Economic Assessment Method

Process costs for the estimated annual production of 2 million tons of cement at the Lehigh Cement plant were considered in the techno-economic assessment. It evaluates the leading economic indicators, such as capital and cement production costs, including raw material, energy, property, plant, and equipment. Figure 3 is a summary table and pie chart for TEA indicators showing the distribution of overhead costs for the annual production of 2 million tons of ordinary Portland cement using coal for thermal energy. Equipment cost estimation was conservatively deduced from the acquisition cost of machines without accounting for freight, installation and taxes, and other capitalizable costs. Local pricing of production materials was used where available and national averages were adopted in other inputs. For instance, 0.1396 USD/kWh, the average electricity price in Maryland [52], was adopted for electricity (see electricity cost sensitivity analysis in Section 3). In contrast, coal and dried biosolids' national average sales price was used.

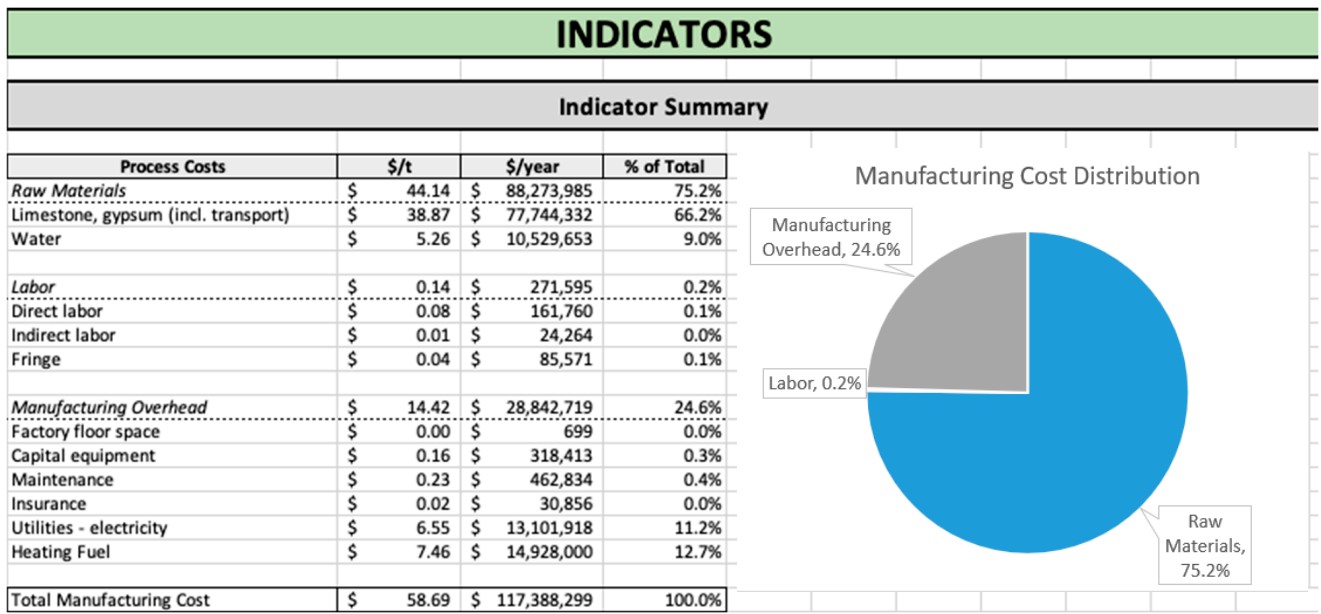

**Figure 3.** TEA Indicator estimates for annual production in the "OPC + Coal" scenario.

Using actual vendor quotes is recommended for TEA accuracy. However, ballpark estimates for TEA parameters such as energy and raw material pricing are sufficient for hotspot analysis and for generating order-of-magnitude estimations [53]. It should be noted that this study is focused on demonstrating the implementation of the continuous assessment and improvement framework; practitioners that adopt this framework for business decisions are encouraged to use actual vendor-provided quotes for their implementation.

### 2.3. Continuous Assessment and Improvement Deployment Framework

Secured deployment of IoT-enabled solutions in the cement industry is drawing the attention of stakeholders because of the significant value in cost reduction, the increased efficiency, and the greater visibility that IoT devices can provide. IoT-related technologies have been explored in the following areas with trials in the cement industry:

- Secured deployment [54];
- Event tracking in supervisory control and data acquisition system [55];
- Fuzzy-logic-based flame image processing for rotary kiln temperature control [56];
- IoT-regulated moisture sensor [57];
- Real-time carbon dioxide monitoring based on IoT cloud technologies—MQ135 carbon dioxide sensor, ESP8266 Wi-Fi module, Firebase cloud storage service, and Android application [58].

Figure 4 is a simplified depiction of the use of connected devices in the production of Portland cement through the dry method. Sensors measure raw materials from the quarry for quality, moisture, and pH value. Transportation of raw materials to the plant is tracked to measure cost and related emissions. Energy requirements of the crushers, fuel consumption, and $CO_2$ emission due to calcination at the kiln, moisture level, temperature, and coolers and grinders' energy consumption are all measured and then transmitted to cloud-based servers through a gateway device.

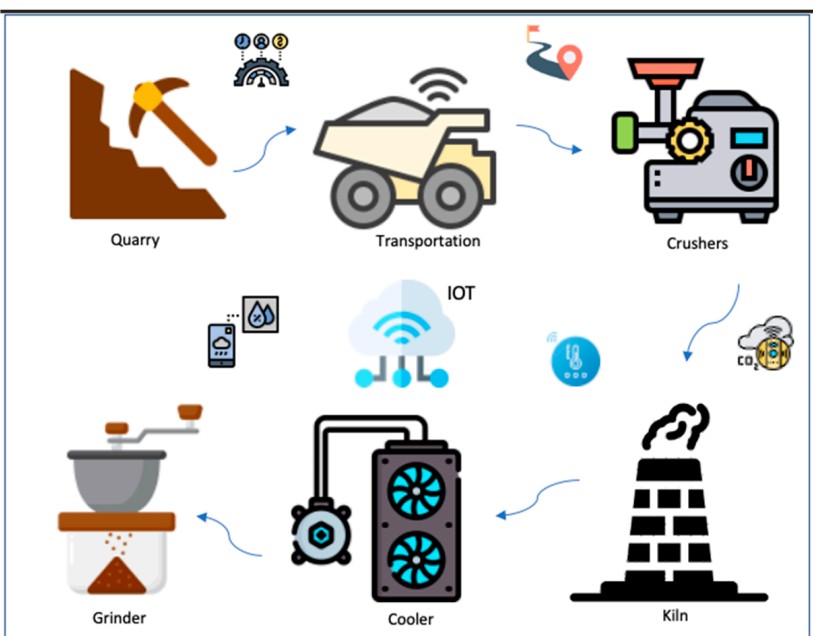

**Figure 4.** IoT-enabled cement production.

CCUS pathways that need early-stage economic and environmental performance assessment template development were identified in the work of [15]. The authors synthesized existing guidelines and approaches into actual templates that can be adopted for early-stage LCA and TEA. These templates are editable with a programming language with Excel handling libraries to enable database integration, sensitivity and uncertainty analysis, and advanced visualizations for decision-making. Python programming language has several open-source libraries for Excel and can be adopted for integrating LCA and TEA templates [59]. In addition, the required scripting, macros, and user-defined functionalities are available in the free version of the open-source Python library Xlwings, which also supports Numpy arrays, Pandas Series, and DataFrames on Windows and macOS.

Figure 5 shows the schematic diagram of a continuous assessment and improvement framework for analyzing the environmental and economic impact of a cement production process improvement and feeding back data-informed decisions for managing the plant. In this framework, data is transmitted from the IoT-enabled cement plant to the cloud data platform via gateway devices that connect the disparate networks and translate communications from one protocol to another, thus allowing bidirectional data flow between the cloud and the IoT devices at the cement plant.

The cloud services include computing infrastructure, platforms, and cloud-native applications that facilitate data flow from IoT sensors to storage and data processing platform. Integration with a data warehouse is seamless with connectors and APIs, giving the platform access to Life Cycle Inventory databases and other economic and operations data required for analysis in near real-time or batch processes. Relevant metrics measured from the cement plant are inserted into the LCA and TEA templates with Python installed on the platform. Similarly, the metrics are fed into sensitivity analysis and data visualization with Python. Economic and environmental improvement insights drawn out of these are

used for business decisions, reporting for compliance, and as feedback for improving the production process.

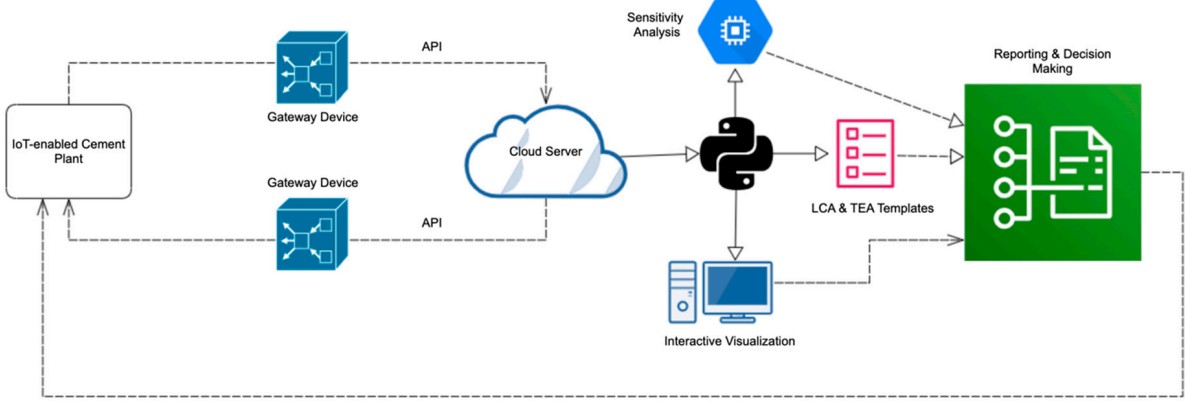

**Figure 5.** Continuous assessment and improvement framework.

## 3. Results and Discussion

### 3.1. Impacts Analysis

Assessing the impact of the significant drivers of $CO_2$ emission for the four scenarios highlights the possible carbon reduction potentials of materials substitution from producing Portland-limestone cement to replace ordinary Portland cement and adopting dried biosolids as a thermal energy alternative to coal. Figure 6 shows the impact assessment results from the "OPC + Coal" scenario, estimating the contributions of the major drivers to global warming potential when using coal for thermal heating to manufacture ordinary Portland cement at the plant. Carbon dioxide ($CO_2$), methane ($CH_4$), and nitrous oxide ($N_2O$) are the relevant gases related to the greenhouse effect. The relative contribution of $CO_2$ is between 98.8% and 100% because it is emitted in much higher quantities than other gases. Based on this impact assessment, calcination leads to more than half of process emissions (about 54%), with the emissions attributable to combustion comprising most of the remaining emissions burden (about 40%). The emission from the other drivers, such as electricity, water, and transportation, contribute an estimated 6% of process emissions.

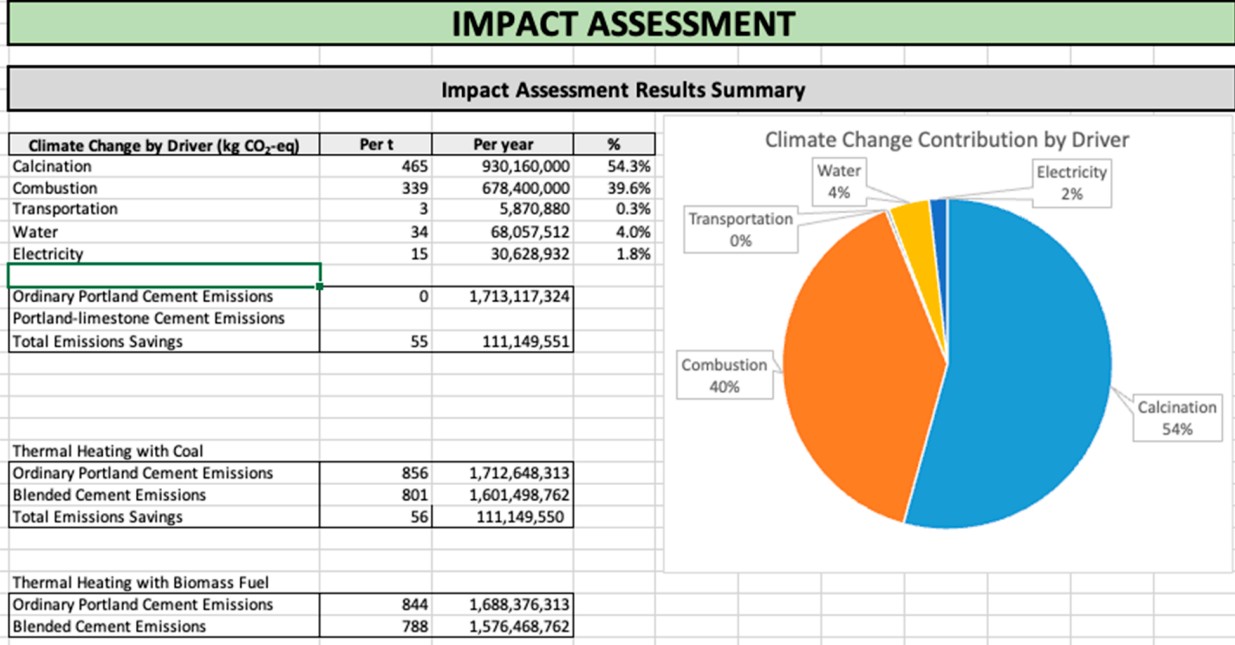

**Figure 6.** Summary and pie chart for LCA impact assessment.

The production of a ton of ordinary Portland cement with thermal energy from coal is estimated to result in a Global Warming Potential (GWP) of 856 kg $CO_2$-eq. A 100-year time horizon GWP, as provided by Intergovernmental Panel on Climate Change in the AR5 Climate Change 2013 report [60], gives a standard unit of measure of how much energy the emissions of a ton of a gas will absorb over some time relative to the emissions of a ton of carbon dioxide, the gas used as a reference.

The estimate of climate change contribution by drivers varies under different scenarios, such as the production of Portland-limestone cement and the use of dried biosolids. Table 4 summarizes the climate change driver contributions in these scenarios. The results indicate the potential of reducing greenhouse gas emissions from 856 kg $CO_2$-eq per ton of ordinary Portland cement using coal for heating to 788 kg $CO_2$-eq per ton of Portland-limestone cement using dried biosolids for heating. This is an opportunity for a 7.9% reduction in $CO_2$ emission from material efficiency.

**Table 4.** Summary of climate change driver contributions for four raw material scenarios.

| # | Scenario | Calcination | Combustion | Others | GWP (kg $CO_2$-eq) |
|---|----------|-------------|------------|--------|---------------------|
| 1 | OPC + Coal | 54.3% | 39.6% | 6.1% | 856 |
| 2 | OPC + DBS | 55.1% | 38.7% | 6.2% | 844 |
| 3 | PLC + Coal | 51.3% | 42.4% | 6.4% | 801 |
| 4 | PLC + DBS | 52.1% | 41.1% | 6.5% | 788 |

OPC = ordinary Portland cement; DBS = dried biosolids; PLC = Portland-limestone cement; GWP = Global Warming Potential; GWP 100 = 100-year time horizon GWP as provided by the IPCC 2013 Fifth Assessment Report [59].

Portland-limestone blended cement, which has additional limestone used as an ingredient amounting to about 15% of the mass, has reduced calcination and fuel combustion $CO_2$ emissions when compared to ordinary Portland cement. The assessment indicates that the production of the blended cement has a 6.4% lower carbon footprint than the production of the ordinary Portland cement at the Lehigh Cement plant at Union Bridge, reducing emissions from 856 kg $CO_2$-eq/t to 801 kg $CO_2$-eq/t. Going by annual production of 2 million tons of cement, the switch to the Portland-limestone cement positions the plant to avoid approximately 123,000 tons of carbon dioxide emissions annually. This could be a cost in excess of USD 3 million if there were a carbon tax of 25 USD/ton. In 2022 carbon tax rates in the United Kingdom were 24 USD/ton and as high as 137 USD/ton in Uruguay [61]. Using renewable energy sources for electricity to power the crushers and grinders would further reduce the climate change arising from process electricity use.

Similarly, using dried biosolids as the thermal energy source instead of coal lowers the carbon footprint by 1.4%, reducing emissions from 856 kg $CO_2$-eq/t to 844 kg $CO_2$-eq/t. The reduction in carbon footprint is from reduced combustion emissions. Combustion of 1 ton of coal produces about 2 tons of $CO_2$, whereas incineration of 1 ton of dried biosolids produces about 1 ton of $CO_2$. Approximately 1.5 times the quantity of well-dried biosolid is required to produce the thermal energy equivalent from coal. However, we need to be circumspect when interpreting data for impact categories because emerging LCA impact categories and inventory items are still under development, can vary depending on the source of data and specific situations in the analysis, and can have high levels of uncertainty that preclude acceptance pending further development.

Using a continuous emission monitoring system (CEMS) is vital for emission data input in implementing the continuous assessment and improvement framework. CEMS is the equipment required to determine the gas concentration or emission rate. It is required for some United States Environmental Protection Agency regulations for continual compliance or to determine if emission standards are exceeded. CEMS uses pollutant analyzer measurements and a conversion equation, graph, or computer program to produce results in units of the applicable emission limitation or standard [62].

*3.2. Economic Analysis*

The economic analysis is premised on the annual production of 2 million tons of cement at Lehigh Cement's Union Bridge plant. Based on the indicator summary, the estimated optimal cost of producing a ton of ordinary Portland cement using coal is USD 58.69. However, producing similarly performant Portland-limestone cement using dried biosolids can reduce this cost by 10.87% to 52.31 USD/ton (see the 'Indicators' and 'Inventory' sheets of the quantitative data spreadsheet in Supplementary Materials).

The sensitivity analysis of the production volume of cement at the plant indicates that given the estimated data points, the cost-optimal production volume is 2.4 million tons at the cost of 58.6 USD/ton. Figure 7 plots the manufacturing cost sensitivity of ordinary Portland cement to the production volume at the plant. The inflection point on the cost is at an annual production volume of 800,000 tons.

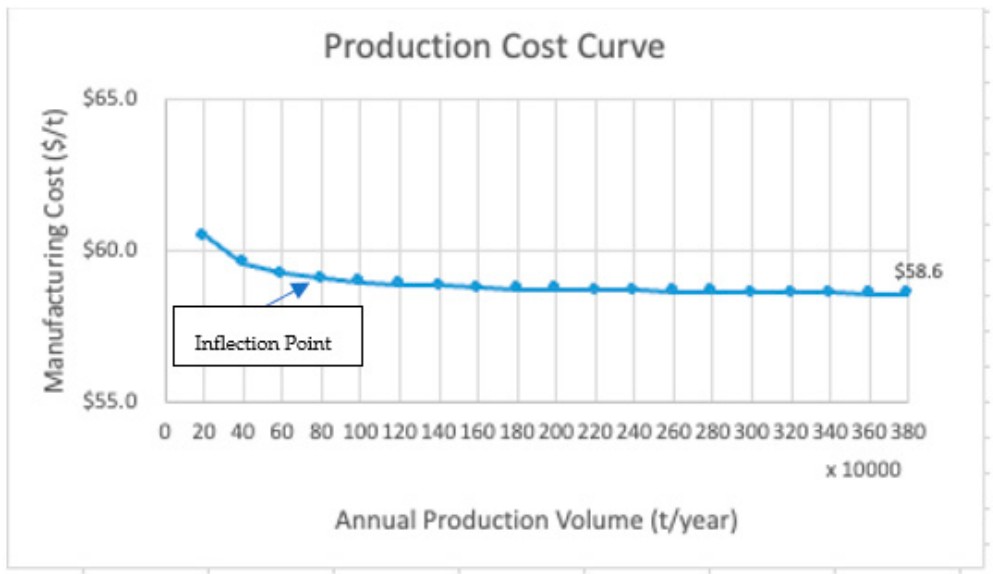

**Figure 7.** Cost sensitivity to production volumes.

In addition, sensitivity analysis of the cost of electricity indicates that production cost increases by 12% to USD 65.77 when the cost of electricity goes up from 0.15 USD/kWh to 0.33 USD/kWh. In September 2021, the average cost of electricity in US states ranged from USD 0.10 in Arkansas to USD 0.33 in Hawaii [63]. Figure 8 is a plot of manufacturing cost sensitivity to the cost of electricity. It indicates that the relationship between the cost of manufacturing and the cost of electricity is linear. The higher cost of electricity increases the cost of manufacturing linearly. Distance from the quarry to the plant can also impact the cost of production due to the increase in transportation costs. However, the cost in this analysis is premised on the proximity of the New Windsor quarry to the cement plant. The crushed material from the quarry is transported to the cement plant via a 4.5 miles-long overland conveyor.

The analysis indicates that the number of personnel available on work shifts did not significantly affect the cost of production because plant personnel cost is only 0.2% of the cost of production. However, fluctuations in the cost of other raw materials can impact the cost of production.

Figure 9 plots the cost sensitivities to production volumes for the four production scenarios highlighted in Table 3. It visually compares the cost sensitivity to production volume using the different production inputs in the four scenarios. The visualization was created using the subplot() function in Matplotlib that enables drawing multiple graphs in a single plot. Matplotlib is a comprehensive Python library for creating static, animated, and interactive visualizations [64].

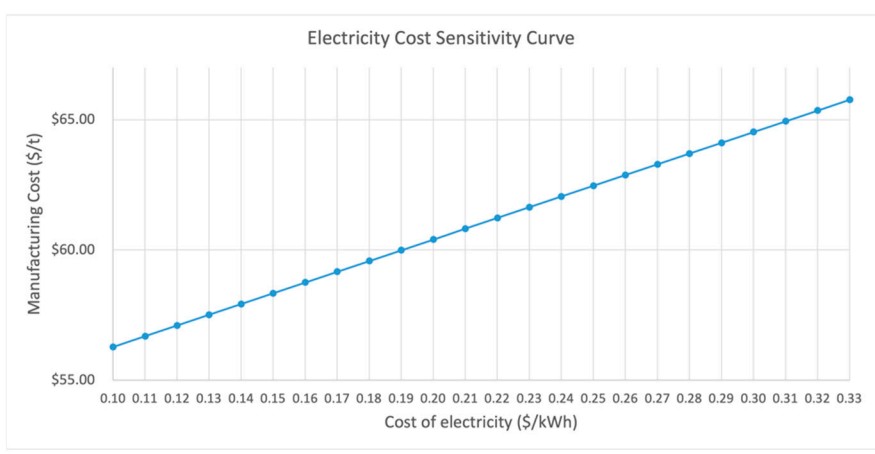

**Figure 8.** Production cost sensitivity to electricity cost.

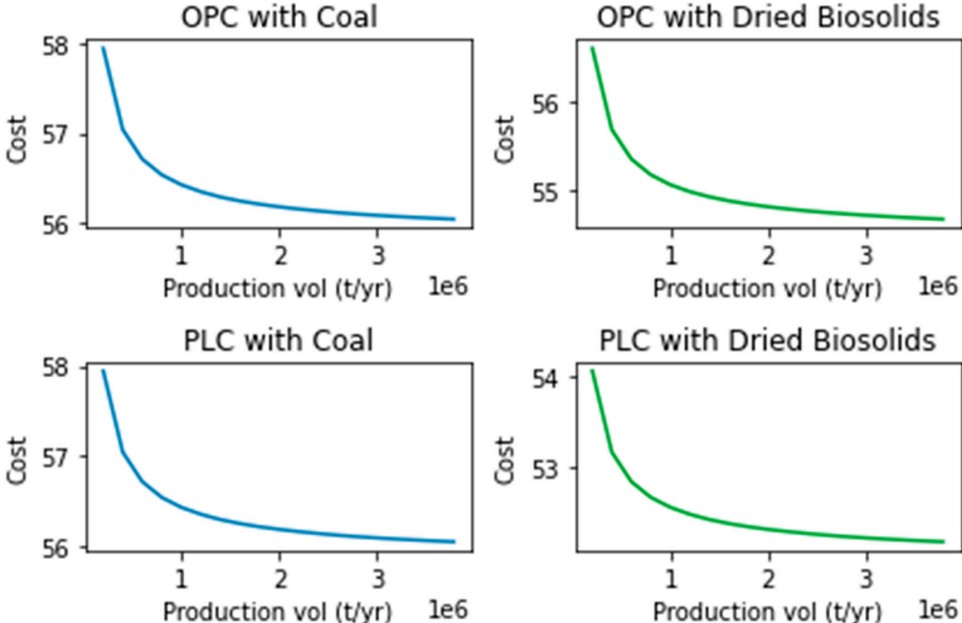

**Figure 9.** Plots of cost sensitivity to production volumes in four production scenarios.

The plots have a similar shape with declining cost/ton as production volume increases until the point of inflection when cost/ton remains stable despite further production increases. Production cost is lower in the scenarios where thermal heat is provided from the combustion of dried biosolids. The model uses fair market pricing for dried biosolids made from drying sewage sludge with a heating value of approximately 16 GJ/t. The economic analysis can be expanded to include the capital investment required to dry the sewage sludge if it is within the boundary of the analysis for a production plant that produces its alternative fuel as part of the circular economic process.

### 3.3. The Deployment Framework

This study contributes to developing practical approaches to integrating sustainability into cement production. It addresses integrating economic and environmental impact assessments into cement production operations. The framework can facilitate the dynamic implementation of sustainability innovations and the continuous measurement of their performance. Beyond mere rhetoric, cement manufacturers that have made a public commitment to fighting climate change can leverage this deployment framework in whole, or parts, to be on their way to meeting their emission reduction targets. This is demonstrated using production scenarios at a US cement plant.

The continuous assessment and improvement deployment framework enables dynamic management of production inputs to optimize the cost of production and reduce carbon emissions. The problem is expressed as a minimization of production cost by defining it as a linear function subject to linear constraints. The constraints considered include the demand for cement types and the cost and availability of raw materials, such as alternative fuels, emission compliance requirements, and the company's commitment to lower emissions. The computation is executed as a linear programming problem to find the feasible region and optimal solution. Data to define the linear constraints based on cost, availability, operational policies, and compliance are pulled from readings at the plant and supplied interactively by process managers. Complex computations and graphical representations of the information are enabled by leveraging appropriate libraries and APIs in the cloud platform. Insights thus generated aid decision-making and positioned the cement plant as a dynamic, innovative, and responsible business committed to its operations' sustainability and open to rapid testing and experimentation.

The framework has opportunities for further optimization. It can also be coupled with other business operation and reporting systems to source actual operations and financial metrics as input data and directly supply needed information to the other systems. Business uncertainties such as raw material price volatility can be quantified using Python libraries for Monte Carlo simulation.

### *3.4. Limitations*

Emerging LCA impact categories and inventory items are still under development, can vary depending on the source of data and specific situations in the analysis, and can have high levels of uncertainty that preclude acceptance pending further development. There is, therefore, a need to be circumspect when interpreting data for impact categories. The accuracy of the results from the LCA and TEA and the correct interpretation of it depends on the accuracy of the data inputs into the templates adopted in this study. Apart from the data sourced from the cement plant, the authors were careful to use data from verified libraries, articles, and data sources.

The LCA in this study is limited to the use phase, often called 'cradle-to-gate'. It only covers the carbon impact of the product from the beginning of production until the product leaves the producing company. In addition, as depicted in Figure 1, the exchanges with the system boundaries in consideration are limited to energy, particulate emissions, gaseous emissions, and heat.

### 4. Conclusions

While the negative effect of climate change is well-known, and it is generally accepted that humanity needs to make changes to slow down climate change intentionally, there are still opportunities to fast-track this intended change with global, national, and local policies that help reduce GHG emissions from human activity. A part of this will be well-defined and standardized methods for measuring and efforts to improve the economic and environmental impacts of activities such as cement production, which are known to result in significant GHG emissions. LCA and TEA templates exist for this purpose, and outstanding metrics that are difficult to measure directly can be estimated with Life Cycle Inventory metrics from relevant databases. From the reviews undertaken in this study, the industry will benefit from continuously measuring the environmental and economic impact of cement production processes and drawing out data insights that will help improve these impacts. The potential reduction in carbon footprint found from material substitutions at the Lehigh Cement plant explored in this study falls within the range published for Portland-limestone by cement manufacturers [65] and in studies on blended cement [66].

Finally, this paper demonstrates a deployment framework for the continuous assessment and improvement of a cement production process's environmental and economic impact. The framework is inspired by implementing IoT at the cement plant and running continuous improvement analytics on the data measured through sensors at the plant and

the data from cement production operations and other integrated databases warehousing relevant Life Cycle Inventory data. The LCA and TEA Supplementary Materials are available in this repository referenced [67]. The insights generated via the analytics are used to improve the cement production operation by providing data points for decision-making. This framework can be used to rapidly measure the impact of various alternative inputs, alternative fuel sources, and carbon sequestration methods suggested by research as they are tried out at the cement plant. It is expected that the insights on environmental and financial trade-offs will help make informed decisions on improvements in cement production and have the potential to contribute to sustainability in cement manufacturing.

**Supplementary Materials:** The following supporting information can be downloaded at: https://osf.io/5kcyp/?view_only=8cbfc104711c4b91b551a7579e054080 last accessed on 8 August 2023; Data spreadsheet titled "Data_Sheet_quantitative.xlsx" containing LCA and TEA worksheets from which the following were extracted—Figure 2: Subsection of the Inventory sheet in the LCA, Figure 3: TEA Indicator estimates for annual production in the "OPC + Coal" scenarios, Figure 6: Summary and pie chart for LCA impact assessment, Figure 7: Cost sensitivity to production volumes, Figure 8: Production cost sensitivity to electricity cost, and Table 4: Summary of climate change driver contributions for four raw material scenarios.

**Author Contributions:** Conceptualization, O.O. and S.S.; methodology, O.O. and S.S.; formal analysis, O.O. and S.S.; writing—original draft preparation, O.O.; writing—review and editing, S.S.; visualization, O.O.; supervision, S.S.; funding acquisition, S.S. All authors have read and agreed to the published version of the manuscript.

**Funding:** The APC was funded by Colorado State University's Systems Engineering Department.

**Data Availability Statement:** Data are contained within the article and Supplementary Materials. The data presented in this study are available at: https://osf.io/5kcyp/?view_only=8cbfc104711c4b91b551a7579e054080 accessed on 8 August 2023.

**Acknowledgments:** We wish to show our appreciation to Kwaku Boakye at the Lehigh Cement plant at Union Bridge, Maryland, who facilitated access to cement plant production information, power consumption data, and context.

**Conflicts of Interest:** The authors declare no conflict of interest.

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
