# Peer review of "Continuous Assessment of the Environmental Impact and Economic Viability of Decarbonization Improvements in Cement Production"

_resources, doi:10.3390/resources12080095_

Round 1

Reviewer 1 Report

Dear Autors,

thank you for your article, which i found both interesting and relevant.

Unfortunately, I cannot consent to a publication yet. See my direct comments below and some additional comments in the pdf-version (highlighted in yellow).

There are 5 issues that need to be addressed in my opinion. These relate to your Research Design, the introduction of the Deployment Framework, the Results for DBS, the missing section on Limitations, and missing information in your Conclusion.

kind regards and good luck

1.  Major Revision

The following points need to be addressed for my consent to a publication, by either solving the issues or rebutting them.

a)   Research Design

It is not entirely clear, which research questions are addressed by the study. Although your introduction provides a good job in guiding the reader towards that end, the actual goal of the study is not spelled out adequately.

b)   Deployment Framework

This exercise is sometimes deemed relevant, and sometimes it looks like an afterthought. Either way, it is not properly introduced methodologically, and it is certainly not discussed properly in terms of results. I suggest one of the following solutions:

(1) Drop this part of the study altogether and focus on the environmental and economic results. You can then introduce this as part of the outlook in your conclusion.

(2)    Introduce a case-study, in which the reader understands how it works and what results one gets under different conditions.

(3)    Introduce, explain, and discuss the equations. For example, which part of the system would be optimized in regard to which parameter changes?

c)   Results

You introduced a table with the main results from your LCA assessment and you discuss how PLC reduced both calcination and combustion emissions. However, you should also discuss how the use of DBS affected the results and why.

d)   Limitation

There is no discussion on the limitations of your approach. Please introduce a chapter, or sub-section, discussing the influence of data quality, assumptions, system boundaries and so on. What are the weaknesses of your approach for each of the methods?

e)   Conclusion

Please compare your results to literature. Given the same or similar system boundaries, are your results in range with other literature sources? Did you simplify the approach, or did you come to new insights? If no literature can be found; how should your results scale in a global context?

2.  Minor Revision

The following points could be considered by the authors in order to improve the quality of the study.

f)    General Comments

·         The authors might want to improve the readability of their study by using shorter sentences and introducing synonyms or explanations for non-common terms (e.g., “templazitation”).

·         Consider adding a list with abbreviations and units at the beginning of your paper

g)   Introduction

The introduction is well written and provides a comprehensive summary of the subject. I have two minor points the authors might want to consider for their re-submission.

·         p.2.: The authors mention that process- and fuel-related emissions “account for a significant portion of total direct emissions”.
I assume from the context that this relates to GHG (or GWP 100a as a derivative). It would be helpful to the reader to include estimations on this portion from literature (maybe in form of a range)

·         p.3.: The authors summarize their study goal in the last paragraph.
Although the reader gets the “gist” of the study, a bit more details are needed. Please introduce

o   which “alternative manufacturing inputs” are considered (all of the above?)

o   what is meant by “deployment framework”, as this term seems to indicate some form of method (is it a method? or is it more in line with some form of roadmap?)

h)   Materials and Methods

The description and discussion of method is sufficient. However, it should include a section, or paragraph, clarifying how the methods are related to each other and to the research question. Some form of research design or framework is needed so the reader understands how and why the authors approached the subject. Further comments are:

·         2.1.1, paragraph: E seems to indicate energy-input or energy-flows rather than energy-exchange. The same holds for H. Is this intentional?

·         2.1.1, figure 1 a): Please indicate if transports are cut-off and if there are other parts in the cradle-to-gate boundaries that are cut-off

·         2.1.1, figure 1 b): Please check whether P is actually PE (which I assumed) or something else

·         2.1.3: Please clarify what impact assessment method is used and how it relates to the changes you investigate; esp. in regard to the carbon accounting of dried biosolids

·         2.2, Table 3: Please check the table for mistakes (e.g., spelling, additional lines) and consistent content (e.g., table-head in regard to “other methods”)

·         2.2.1: Please discuss how the conservative cost estimation affects your results later on

·         2.3: The last paragraph seems to indicate the goal of the entire exercise; this should be introduced and discussed earlier

i)     Results

·         3.1.: Your reference to GWP 100a should be placed in the method section, which should also include a reference to the impact method used and how this method deals with biotic carbon.

·         3.1.: Your results for PLC are interesting. Any idea or literature suggestion why this is the case?

·         3.2: You mention a “uncertainty analysis” as well as a “sensitivity analysis”. These are fixed terms in LCA methodology but can refer to different methods. Please introduce what type of UA and/or SA was conducted, and which parameters were changed and why.

·         3.2, Figure 8: This figure seems superfluous in its current design. Consider adding meaningful data to it or show why it is necessary here.

Reviewer 2 Report

Excellent research relevant to be published and spread

Author Response

Thank you for taking the time to review it.

Reviewer 3 Report

This manuscript provides a continuous assessment of the environmental impacts made by the cement industry and viable solutions to contribute in decarbonization of the climate. Overall manuscript is written very well, and it provides a great basis for the cement industry to contribute in the decarbonization.

I have no comments further. 

Good luck to authors. 

Author Response

(The authors gave the same response as above.)

Round 2

Reviewer 1 Report

Dear Authors,

After reading your responses and corrections, I can give my consent for a publication.

kind regards

Reviewer